# FeZrN Films: Magnetic and Mechanical Properties Relative to the Phase-Structural State

**DOI:** 10.3390/ma15010137

**Published:** 2021-12-25

**Authors:** Elena N. Sheftel, Valentin A. Tedzhetov, Eugene V. Harin, Philipp V. Kiryukhantsev-Korneev, Galina S. Usmanova, Olga M. Zhigalina

**Affiliations:** 1Baikov Institute of Metallurgy and Materials Science RAS, Leninsky Prospekt 49, 119334 Moscow, Russia; sheftel@imet.ac.ru (E.N.S.); harin-eugene@ya.ru (E.V.H.); gusmanova@imet.ac.ru (G.S.U.); 2Department of Materials Science, Moscow Aviation Institute (National Research University), Volokolamskoye Shosse 4, 125993 Moscow, Russia; 3Department of Powder Metallurgy & Functional Coatings, National University of Science & Technology “MISIS”, Leninsky Prospect 4, 119049 Moscow, Russia; 4Laboratory of Electron Microscopy, Shubnikov Institute of Crystallography of FSRC “Crystallography and Photonics” of RAS, Leninsky Prospect 59, 119333 Moscow, Russia; zhigal@crys.ras.ru; 5Department of Material science, Bauman Moscow State Technical University, 2nd Baumanskaya Street 5, 105005 Moscow, Russia

**Keywords:** Fe-based films, magnetron sputtering, nanocrystalline, phase composition, transmission electron microscopy, X-ray diffraction, nanoindentation, magnetic properties

## Abstract

The paper presents results of investigation of Fe_65.3–100_Zr_34.7–0_N_7.5–0_ films prepared by dc magnetron deposition on glass substrates and subsequent 1-hour annealing at temperatures of 300–600 °C. The influence of the chemical and phase compositions and structure of the films, which were studied by TEM, SEM, XRD, and GDOES, on their mechanical properties determined by nanoindentation and static magnetic properties measured by VSM method is analyzed. The studied films exhibit the hardness within a range of 14–21 GPa, low elastic modulus (the value can reach 156 Gpa), and an elastic recovery of 55–83%. It was shown that the films are strong ferromagnets with the high saturation induction *Bs* (up to 2.1 T) and low coercive field *Hc* (as low as 40 A/m). The correlations between the magnetic and mechanical properties, on one hand, and the chemical composition of the films, their phase, and structural states as well, on the other hand, are discussed.

## 1. Introduction

The recent tendency to store more and more data (the amount of data being recorded is increasing by 30 to 40% per year) and for a longer time, which is necessary for archiving, backup, and data protection, cannot be realized by increasing the number of disks since energy expenses related to the complex manufacturing technology of the disks and larger spaces for the storage of them increase their total cost. Tape, which is the storage of magnetic bits on flexible 0.5” wide media, provides critical data storage needs as evidenced by continued growth in exabyte shipments of tape products [1,2,3,4,5,6,7]. Since the invention of the tape, the recording density on the tape has risen by six orders of magnitude due to the increase in the coercive field of media [6,8]. Future improvements in tape areal density and tape cartridge capacity are expected to be achieved from evolutionary developments in the tape-head technology, that is, not requiring revolutionary developments, such as energy assisted magnetic recording technologies required by HDD strategies [2,6].

The development of the tape-head technology largely depends on the success in designing the soft magnetic films with a unique combination of properties required for the magnetic head applications. The combination includes the high saturation induction (which provides a high magnetic field at a high-coercive medium during signal recording), low coercive field (which provides a low-field head saturation) and high wear resistance (a medium-head contact couple works under wear conditions). As for the wear conditions, the recording head should be located near the tape surface, since the amplitude of the recording signal improves with decreasing the head-to-medium distance. This leads to the wear of heads, defects on their working surface, and additional costs for the replacement and restoration of the heads. Thus, the development of film alloys meeting the combination of properties required for the application of the alloys in storage devices with tape-head technology is an actual and significant problem. Note that the use of magnetic materials for power electronic applications, in particular in inductors and transformers [9], requires the same properties of magnetic materials, namely, the low coercive field, high saturation induction, and high resistivity for high magnetic permeability in rf magnetic fields [10,11,12,13], which allow the size of the converter, while maintaining the high current, to be reduced [14].

The revolutionary jump in the magnetic recording density, which took place in 1990s, was determined by technological advances, in particular, related to the development of new highly soft-magnetic nanocrystalline alloys, such as FINEMETs (FeCuNbSiB) applied in heads. In the early 1990s, nanocrystalline soft-magnetic alloys of the Fe-Me^IV,V^-X (X = B, C, N, O, and Me^IV,V^ = Ti, Zr, Hf, Nb, Ta, etc.) systems came to the attention for the head application. The alloys in the form of films are prepared by magnetron sputtering technique. It was supposed that the structure of these films consisting of two nanocrystalline phases (main ferromagnetic αFe-based phase and a thermodynamically stable nonferromagnetic MeX compound) is capable to provide the magnetic properties which are higher than those of FINEMETs. As for the films of the Fe-Zr-N system [15,16,17,18,19,20], the published works in fact are devoted to the study of the Fe-N system films and the influence of small Zr additives on their structural and magnetic properties. It was shown that the small Zr additives affect the redistribution of N between grains containing the saturated solid solution αFe(N), grain boundary areas, and iron nitrides, the size of crystalline phase grains, micromagnetic structure, some magnetic properties of the Fe-N films with various nitrogen contents. It was shown that the soft-magnetic properties of Fe-N films get better, their structural and soft magnetic stability is improved with small Zr additives owing to, in particular, the fact that Zr reduces the magnetostriction of Fe and by the presence of Zr in grain boundary area.

Since the early 2000s, the authors of the presented work have been carried out studies of the Fe-Me^IVa^-X system soft-magnetic films (Fe-Zr-C, Fe-Zr-N and Fe-Ti-B) [21,22,23,24,25,26,27,28,29,30,31], wherein the large amount of systematic and comprehensive experimental data on Fe-Zr-N films were obtained [24,25,26,27,28,29,30,31]. The studied film compositions were chosen using the physical-chemical approach to the alloy design in order to ensure the formation of a two-phase structure (αFe-based phase + MeX) under appropriate preparation conditions of the film [32].

To date, there are no data on the mechanical properties of the soft-magnetic FeZrN films; therefore, the aim of this work is to study the effect of chemical and phase-structural states on the mechanical and magnetic properties of FeZrN films prepared by dc magnetron deposition.

## 2. Materials and Methods

For the study, films of six series I, II, III, IV, V, and VI were prepared (Table 1). The calculated composition was supposed to produce films containing about 10 mol.% ZrN (films III), which are characterized, according to our earlier data [24,33], by the high saturation induction *Bs* and low coercive field *Hc*; films II, IV, V with a lower Zr content, as compared to that in films III, and the ratio at.% N/at.% Zr = 1–2 were taken to obtain a higher saturation induction *Bs* as compared to that of films III; Fe films (I) and films with a high Zr content (VI) were taken to obtain diffraction patterns from the material in crystalline and amorphous state, respectively.

The films were deposited onto glass and metallic (Ni-Cr alloy) substrates by dc reactive magnetron sputtering of a heated Fe_1-x_Zr_x_ composite target in Ar, Ar + 5% N_2_ and Ar + 15% N_2_ atmospheres (Table 1) and subsequent vacuum annealing (at the residual pressure *p* = 2.4–4.2∙10^−4^ Pa) at 300, 400, 500, and 600 °C, exposure 1 h. The films as-sputtered and annealed at 400, 500, and 600 °C (excluding films IV that were annealed at 400 and 500 °C) were studied. In terms of the present work, the choice of the annealing temperatures equal to 400, 500, and 600 °C is based on the results of our previous works [32,33,34], in which it was shown that the magnetic properties (*Bs* and *Hc*) of the as-deposited Fe-Zr-N films change during annealing at the above temperatures. The annealing at 400 or 500 °C (in accordance with the film composition) provides the best combination of the film soft magnetic properties that begin to get worse at 600 °C due to the diffusion interaction of the film with the substrate. As for the films I (Fe), they were obtained only in as-deposited state for the study of the influence of Zr and N additions on structure and properties of Fe films. It is obvious that annealing of as-deposited Fe films does not improve their properties.

The target was a Fe disk, with Zr chips evenly distributed in the erosion zone, the content of which was ranged from 0 to 13.4 wt.% of the entire target (the information on the preparation method is available in detail in [27]). The deposition time of all studied films was 7 min. The chemical composition of the films on glass substrates was determined by energy dispersive X-ray spectroscopy (EDX) using a Hitachi S-3400N (Hitachi High-Technologies, Tokyo, Japan) scanning electron microscope (SEM) equipped with a Noran 7 Thermo attachment. The film thickness (Table 1) was estimated using the cross-sectional electron micrographs (Figure 1a) (SEM). An accurate assessment of the content of light elements (N, O) was carried out for the films deposited on metallic (Ni-Cr) substrates by glow discharge optical emission spectroscopy (GDOES) using a Horiba Jobin Yvon Profiler 2 (Horiba Jobin Yvon, Longjumeau, France). The Ni-Cr substrates were used exclusively for GDOES measurements on the as-deposited films. The rest of the measurements were performed on the glass substrates.

The analysis of the phase and structural states (the volume fraction *v_i_*, lattice parameter *a*, and grain size *D* of phases and microstrains *ε* in grains) was carried out by X-ray diffraction (XRD) analysis performed on a Rigaku Ultima IV diffractometer (Rigaku Corporation, Tokyo, Japan) using CuKα radiation and the full-profile Rietveld method [27,35]. The lattice parameters of the phases were calculated from the position of the gravity centers of the high-angle reflections using the Voigt approximation. The grain size was taken to be equal to the coherent domain size. Transmission electron microscopy (TEM) and electron diffraction (ED) analysis were performed using a Tecnai G2 30ST electron microscope (FEI Company, Hillsboro, OR, USA) operating at an accelerating voltage of 300 kV. Cross-sections and “in-plane” samples for the electron microscopic studies were prepared in a Gatan PIPS 691 (Gatan Inc., Pleasanton, CA, USA). Thinning was carried out using special pastes, sprays, as well as argon ions with an energy of 3–5 keV and an angle of incidence of 3–5 degrees using a Gatan PIPS 691. Images taken with the electron microscope were processed and analyzed using the Digital Micrograph and TIA software.

The mechanical properties of the films (hardness *H*, elastic modulus *E*, and elastic recovery *W*) were studied by the continuous nanoindentation method using a third-generation NanoTest measuring system (Micro Materials Ltd., Wrexham, UK) in accordance with the standard ISO-14577. During the tests, a Berkovich diamond indenter (included with a NanoTest) was used. The analysis of the indentation results was carried out according to the Oliver–Pharr method (by mathematical processing of the descending curve on the *P-h* diagram) [36].

Static magnetic properties (the saturation induction *Bs* and coercive field *Hc*) were measured in applied fields up to 1280 kA/m using a LakeShore 7407 vibrating-sample magnetometer (Lake Shore Cryotronics Inc., Woburn, MA, USA). The measurement error of *Bs*, which is related to the difference in the shape and size of a standard nickel ball (3 mm in diameter) and samples, does not exceed 10%. The vibrating sample magnetometer uses a Hall sensor (included with a LakeShore 7407) to measure the magnetic field in the gap of an electromagnet regardless of the current strength and remanence of the electromagnet. The hysteresis loops are built according to the Hall sensor readings, and the current in the electromagnet is used only to correlate the Hall sensor readings with the measured magnetic moment. Note that all measurements started with a maximum field of 1280 kA/m, then, after passing through the zero field, a negative field of 1280 kA/m was reached and the cycle was closed in reverse order. The accuracy of the magnetic field measuring was no worse than 10 A/m. All measurements were made in the film plane.

## 3. Results and Discussion

### 3.1. Chemical Composition and Morphology of the Films

The chemical composition of the as-deposited films I (Fe), II (Fe_92.2_Zr_3.0_N_4.8_), III (Fe_83.0_Zr_9.5_N_7.5_), IV (Fe_89.6_Zr_3.4_N_7.0_), V (Fe_88.7_Zr_4.4_N_6.9_), and VI (Fe_65.3_Zr_34_) is presented in Table 1 and in the Fe-Zr-N concentration triangle (Figure 2). As is seen from Figure 2, the compositions of the films II, III, IV, V are located near the two-phase αFe + ZrN area of the equilibrium Fe-Zr-N phase diagram (scheme). The scheme (Figure 2) is constructed based on the analysis of the Fe-N, Fe-Zr, and Zr-N binary phase diagrams [37,38,39] and the corresponding Fe-Ti-N system [40]. All films contain impurity oxygen (no more than 2 at. %). The chemical composition given in Table 1 indicates the main elements (Fe, Zr, and N).

The element distribution profiles over the cross-section of the films, which were measured by the GDOES method, indicate uniform distributions of elements across the film depth (Figure 1b). Dark-field images (TEM) of the cross-section of the films (Figure 1d) indicate the formation of the columnar structure directed perpendicular to the film-substrate interface in films II and IV with 3 and 3.4 at.% Zr and the ratio of at.% N/at.% Zr = 1.6 and 2, respectively.

The two layers, the bottom amorphous layer located near the film–substrate interface and the top layer with a crystalline structure, are formed in the films with the highest contents of N (IV, V) or both Zr and N (III), which are the iron amorphizing elements. The amorphous layer thickness increases with increasing N or both Zr and N contents. So, for films IV and III, the amorphous layer thickness is 100 and 200 nm (Figure 1c,d), respectively. According to the EDX data (Figure 1e–g), there is an increased content of Zr in the amorphous layer, relative to the Zr average content in the film, and the gradient of the content along the thickness of the amorphous layer thickness from 13.5 at.% (Figure 1c,g, point 3) to 11.9 (Figure 1c,f point 2) and 9.8 (Figure 1c,e point 1) in the direction from the substrate to the film surface. Note that it is impossible to see the features of the chemical composition in the amorphous layer with a thickness no more than 200 nm in the GDOES curves (Figure 1b), since it is spread over the entire depth (more than 0.5 μm) of the roughness of the Ni-Cr substrate.

### 3.2. Structure and Phase Composition

#### 3.2.1. As-Deposited Films

The full-profile XRD patterns (Figure 3a) taken for films I (Fe) and III (Fe_83.0_Zr_9.5_N_7.5_) exhibit the reflections located at ~44.5°, ~64° and ~82°; for films II (Fe_92.2_Zr_3.0_N_4.8_), the reflections are at ~44.5° and ~82°. All reflections are shifted to the 2*θ* low-angle range relatively to the positions of proper (110), (211), and (200) αFe reflections. As the Zr and N contents increase, the shift of the reflections becomes more substantial. These data mean the presence of a body-centered cubic (bcc) phase with the lattice parameter that exceeds that of pure αFe (2.866 Å). So, the bcc phase lattice parameter is 2.872 Å, 2.904 Å, and 2.895 Å for films I (αFe), II (Fe_92.2_Zr_3.0_N_4.8_), and III (Fe_83.0_Zr_9.5_N_7.5_), respectively (Table 1, Figure 4).

The XRD pattern exhibits, along with strong narrow reflections corresponding to crystalline Fe-based bcc phase, wide reflections strongly diffused and superimposed on crystalline phase reflections centered at 2*θ*~39–40° (Figure 3a). The latter correspond to the amorphous (most likely in terms of XRD) phase. The presence of amorphous phase enriched in Fe is confirmed indirectly by the ratio of the (110) reflection intensity in the XRD patterns taken from the films (Figure 3a) to a 100%-Fe crystalline standard, which is less than unity. The wide reflections had not been resolved in frame of this work.

The ED data are consisted with the XRD data. The selected area diffraction (SAED) patterns for the as-deposited II and III films (Figure 5a,b) indicate weak diffuse rings corresponding to planes (200), (112) and strong diffuse ring of (110) bcc αFe, which testify the formation of a mixed structure (nanocrystalline + amorphous, which, more likely, is amorphous in terms of XRD). Herein, the interplanar spacings (*d*_112_ = 1.17–1.19 Å, *d*_200_ = 1.44–1.45 Å, *d*_110_ = 2.04–2.052 Å) exceed significantly those for αFe (1.17, 1.43, 2.027 Å). Thus, the XRD and ED data show that the single phase formed in the as-deposited films II and III is bcc super saturated αFe(Zr,N) solid solution (it will be indicated as bcc phase).

It is known that the nitrogen solubility in bcc Fe in the equilibrium Fe-N system is negligible, namely, only 0.4 at.% at 590 °C [37]. The phenomenon of the formation of bcc supersaturated solid solution αFe(Zr,N) in sputtered FeZrN films is discussed in our previous works [27,30].

The XRD patterns taken for films IV (Fe_89.6_Zr_3.4_N_7.0_) and V (Fe_88.7_Zr_4.4_N_6.9_) with the low Zr content and the ratio at.% N/at.% Zr ~2 and ~1.6, respectively (Table 1, Figure 3a), exhibit, along with the (110), (200), and (211) reflections corresponding to bcc phase, the additional reflections (001), (111), (002) observed at 2*θ* angles of ~23, ~41, ~47° corresponding to face-centered cubic (fcc) Fe_4_N phase. The ratio of the volume fractions for αFe(Zr,N) and Fe_4_N is 0.58/0.42 (Table 1).

The SAED patterns taken for films IV and V (Figure 5c,d) agree well with XRD data. The rings corresponding to both the bcc phase (*d*_112_ = 1.18–1.19 Å, *d*_200_ = 1.43–1.44 Å, *d*_110_ = 2.01–2.10 Å) and the fcc phase (*d*_202_ = 1.31–1.35 Å, *d*_002_ = 1.9 Å and *d*_111_ = 2.2 Å) are visible in the SAED patterns (Figure 5c,d). As is seen for films V (Figure 5d), some of the bcc phase rings (*d*_110_ = 2.03 Å) are merged with the fcc phase rings (*d*_002_ = 1.9 Å and *d*_111_ = 2.2 Å), testifying a high dispersity of both phases. Then, an increase in ratio at.% N/at.% Zr reflects the formation of two-phase αFe(Zr,N) + Fe_4_N structure in the films with 3–4.4 at.% Zr.

The XRD data show that the bcc phase is characterized by pronounced axial <011> + <211> texture observed in the films II and IV and partly weaker axial <011> + <211> texture in the films III and V; the fcc phase is characterized by pronounced axial <001> texture (Figure 3a). The areas of increased brightness located on the rings corresponding to the αFe(Zr,N) and Fe_4_N phases are clearly visible on the ED patterns taken from films II and IV (Figure 5a,c). It means that the grains of both phases are oriented to their appropriate planes along the film plane.

The XRD pattern taken for the film VI (Fe_65.3_Zr_34.7_) with the highest Zr content (Figure 3) exhibits two wide diffuse reflections located at 2*θ* angles of ~38 and ~65° (Figure 3a). This pattern is typical of an amorphous material. Therefore, Zr is an efficient amorphizer of Fe.

As the Zr and N contents in the films increase, the lattice parameter of the bcc phase increases (Figure 4). One can see from Figure 4 that the lattice parameter of the bcc phase in the two-phase films (IV and V) becomes lower than that in the single-phase films (II and III) at the comparable Zr and N contents in the films. Probably a predominance of kinetic factors over thermodynamic ones during deposition of the two-phase films (IV and V) provides the advantage to the Fe_4_N phase formation in comparison with the bcc phase leading to the depletion of the bcc phase of nitrogen.

The XRD data show that all the films contain crystallites (grains). The size of the bcc phase grains is 2–3.5 nm according to the XRD data for films III and V (Table 1, Figure 4) and less than 10 nm and 20 nm according to statistical analysis of the TEM images of films III, IV, V, and II, respectively (Figure 1c,d and Figure 5a–d). Since the bcc solid solution is enriched in Zr and N, the bcc-phase grain size decreases (Figure 4) as the Zr and N contents increase in the film. As a result, the grain growth is inhibited by the mechanism of solid solution hardening. The very large size, 45.6 nm, of the bcc phase grains in Fe films (Figure 4) indirectly illustrates the influence of Zr and N on the bcc phase grain size. Concerning the two-phase films (IV, V), the second phase Fe_4_N restrains the growth of the bcc phase grain as well.

When a uniform distribution of most crystallites takes place, we can see columnar agglomerates in films II and IV, which tend to be oriented along the film growth direction (Figure 1d). The columns are not basic structural elements, but are agglomerates of identically oriented nanograins [41,42]. As is noted above for films II and IV, the preferential orientation of the grains comprising the columns is confirmed by the increased brightness areas located in the ED pattern rings corresponding to the αFe(Zr,N) and Fe_4_N phases (Figure 5a,c). The analysis of the TEM images (Figure 1d and Figure 5a,c,e,f) indicates that the expression degree of the columnar structure increases under conditions that favor the development of the crystallization process (i.e., with a lower content of amorphizing elements Zr and/or N in the films or the annealing temperature).

It should be noted that the main phases formed in the films are characterized by the high microstrain in the grain: 0.01–0.3% for the bcc phase and 0.2–0.7% for the fcc phase (Table 1).

#### 3.2.2. Annealed Films

As the annealing temperature increases, the following processes take place in the films: (1) the crystallization leading to the increase in the volume fraction of nanocrystalline phases (the bcc phase reflection intensity and physical broadening become higher and smaller, respectively (Figure 3b,c) and formation of the more pronounced columnar structure (Figure 5e,f); (2) the depletion of the bcc αFe phase of Zr and N (the decrease of the phase lattice parameter, Figure 6, Table 1) without changes in the phase grain size up to 500 °C (Table 1, Figure 6); (3) the formation of the additional phases, fcc ZrO_2_ and hexagonal close-packed (hcp) Fe_3_N, in the films V annealed at 600 °C (Figure 3c and Figure 5f); (4) the formation of the additional phases, fcc ZrO_2_ and fcc FeZr_2_, in the films VI annealed at 500 and 600 °C, respectively (Figure 3d); (5) no visible change of the microstrain value in the films annealed up to 600 °C (Table 1).

It should be noted that the formation of the fcc ZrO_2_ phase in films V and VI could be related to both the presence of impurity oxygen in the films and its diffusion from the substrate to the film at 500 and 600 °C [43]. Oxygen, characterized by a high affinity for Zr (one of the highest formation energies of −355 kJ/mol∙atom), with its corresponding impurity content, should have formed an oxide phase both in the as-deposited state and in the annealed state at temperatures below 500 °C. However, the appearance of the ZrO_2_ phase only in the films annealed at temperatures above 500 °C (Table 1) indicates that the formation of this phase is not the result of the influence of impurity oxygen.

### 3.3. Mechanical Properties

The values of hardness *H*, elastic modulus *E*, and elastic recovery *W* of the films are given in Table 2. All films demonstrate the high hardness (13.7–20.8 GPa) in the as-deposited and annealed states (Table 2, Figure 7). As the Zr and N contents in the films increases in the sequence film I (Fe) → film II (3–4.8 at.%) → film V (4.4–6.9 at.%) → film III (9.5–7.5 at.%), the hardness increases from 16.1 to 20.8 Gpa. This is related to the solid solution hardening of Fe with Zr and N. The lower hardness for film IV with the high N content (7.0 at.%) and with the low Zr content (3.4 at.%) indicates the greater effect of Zr, as compared to that of N, on the solid solution hardening. In a similar sequence of the films I → II → IV → V → III, the grain size of the bcc phase decreases (Figure 6), which also makes the contribution to the observed increase in the hardness of these films (Figure 7), according to the Hall–Petch law *H~D^−^*^1/2^ [44,45].

The high hardness of the as-deposited films is also determined by the high microstrain in the grains (Table 1) and high compressive stresses formed in the films during magnetron deposition, as was shown earlier in our work [31]. According to [46], the values of hardness and compressive stresses formed in the films are related by the dependence *H*~−3*σ*. The low hardness 14.8 GPa (Figure 7) of the films VI (34.7 at.% Zr) is explained by their amorphous structure [46].

The annealing of the films leads to the decrease in the hardness of all films, except film VI. Thus, the hardness of the films II and V annealed at 600 °C decreases by 18% and 23%, respectively (Figure 7). This is explained by (1) the change of compressive stresses to tensile stresses of lower values [31], (2) a decrease in the effect of solid solution hardening due to the depletion of the αFe-based solid solution of zirconium and nitrogen (Figure 6), (3) as well as the formation of the columnar structure (Figure 5e,f) [47,48,49]. The annealing of the amorphous film VI at 500 °C leads to the increase in the hardness by ~26% (Figure 7, Table 2) due to the formation of a nanocrystalline structure in the films [46] and, in this regard, the actions of other, in comparison with the amorphous structure, strengthening mechanisms [50].

To achieve the high wear resistance, in addition to high hardness, nanocrystalline films must have a low elastic modulus *E* and high elastic recovery *W* [51,52]. The values of the elastic modulus of heterophase films IV, V in the as-deposited and annealed states are lower (157–166 GPa) than those for single-phase films I, II, III, VI. This is most likely to be due to the high volume fraction (up to 0.57, Table 1) of the Fe_4_N nitride phase, which is characterized by the significantly lower elastic modulus (157–159 GPa) [53] as compared to that of αFe (~208 GPa) [54]. All films have high values of elastic recovery, which vary in a range of 55–83% in accordance with the chemical composition of the films and annealing temperature (Table 2).

### 3.4. Magnetic Properties

By the value of the saturation induction, all films, except for the films VI, are strong ferromagnets (Figure 8a). The films II have the induction close to that of pure Fe (within the measurement error) and, upon annealing, the induction changes only slightly. For the films III–V, the decrease in the induction upon annealing can be associated either with the formation of iron nitrides (the films III), which could not be detected by XRD, or with an increase in the fraction of nitrides and oxides (the films IV and V). The films VI in the as-deposited state and annealed at 300 and 400 °C are amorphous and have the saturation induction less than 0.01 T; after annealing at 600 °C, they have the saturation induction ~0.15 T, which is due to the crystallization of the amorphous phase and the appearance of the ferromagnetic Fe_2_Zr phase.

The coercive field of the films in the as-deposited state and after annealing at 300 and 400 °C (Figure 8b) ambiguously depends on the phase composition, grain size of the ferromagnetic phase and its alloying, since the additional factors of microstrain, residual stresses, and complex symmetry of magnetic anisotropy in heterophase films are superimposed. The films V have the lowest coercive field after annealing at 500 and 600 °C, as they are the most alloyed and have the smallest grain size (Figure 4), which is consistent with the random anisotropy model [55].

Complex symmetry and volumetric distribution of magnetic anisotropy in heterophase films manifest themselves in the shape of hysteresis loops (Figure 9). Measurements of hysteresis loops in different directions in the film plane are in good agreement with each other (not shown in order to Figure 9 was not cluttered up). 

In all films in the as-deposited state, hysteresis loops indicate that a significant part of the volume of each film has strong anisotropy field. This can be caused by the columnar agglomeration of grains (the films II and IV) or by compressive residual stresses (the films I, II, and V) combined with positive magnetostriction of the ferromagnetic phase [56]. The magnetoelastic anisotropy field 3*λ_s_**σ*/*M_s_* (*M_s_* = 1.3 T, *λ_s_* = 10^−5^ is the saturation magnetostriction, and *σ* = 10^9^ Pa is the residual stresses [31]) does not exceed 24 kA/m for the films, and the anisotropy field in all films, except for films I and VI, fits into this range. After annealing, for all films, except for films III and IV, the coercive field decreases and the anisotropy field weakens. This indicates the predominance of the exchange interaction over the local magnetic anisotropy [27]. Note that residual stresses are not the only source of magnetoelastic anisotropy. The magnetostriction itself causes a stress *E**λ_s_* (E is the elastic modulus) of ~1 MPa, which cannot be detected against the background of residual stresses. In addition, the competition between the natural magnetoelastic anisotropy (3/2)*λ_s_*^2^*E* (independent of the presence of stresses) and the effective magnetic anisotropy of the material (includes all possible types of magnetic anisotropy) causes the dependence of the elastic modulus on the magnetization of the material (Δ*E*-effect [57]). Therefore, to reduce the ΔE-effect, it is necessary to reduce the elastic modulus.

The films III have the most complex magnetic anisotropy (Figure 9), which is expressed in the coupling field in the hysteresis loop (the presence of a coupling field holding part of the sample magnetization in one direction) in the as-deposited film and after annealing at 300 and 400 °C. Note that the film VI annealed at 600 °C has a relative remanence of 0.64, whereas the film III annealed at 400 °C has that of 0.07. However, despite such a low remanence, sample III annealed at 400 °C does not show strong anisotropy field, and at the same time, the magnetization and demagnetization curves near the coupling field run almost vertically.

The reason for such a coupling may be the antiparallel alignment of magnetizations in alternating layers of a multilayer film [58], and the effect manifests itself both in the case of magnetization in the film plane [59,60], and during magnetization perpendicular to the plane [61]. Note that this effect manifests itself only in the films III, for which the amorphous layer thickness is highest (about 200 nm) and the entire film thickness is lowest (0.8 μm).

The topological coupling field is analytically expressed in the “orange peel” model [62,63] and is equal to *H_cf_* = (2^1/2^*λt_fe_*)^−1^*π*^2^*l*^2^*M_s_*exp(−2^3/2^*πt_d_*/*λ*) [64], where *l* and *λ* are amplitude and wavelength of roughness at the interface, respectively (in our case, they are equal to the grain size [65], Table 1); *M_s_* is saturation magnetization (1.8 T); and *t_fe_* and *t_d_* are the thicknesses of the ferromagnetic layer and the non-ferromagnetic layer (the “dead” layer on the surface of the ferromagnetic layer does not exceed 1 nm [66]), respectively. After substitution of values suitable for the films III, we obtain *H_cf_* equal to ~240 A/m, which is close to the observed value of the coupling field (Figure 9).

After annealing at 500 and 600 °C, the amorphous layer crystallizes; this leads to an increase in the coercive field, and the anisotropy field increases due to the strengthening of the columnar structure (Section 3.2). The films IV and V also contain an amorphous layer; however, no coupling field is noticeable in their hysteresis loops, probably owing to the several times smaller thickness of the amorphous layer relative to the entire film thickness (Table 1).

## 4. Conclusions

The films of six series, I (Fe), II (Fe_92.2_Zr_3.0_N_4.8_), III (Fe_83.0_Zr_9.5_N_7_), IV (Fe_89.6_Zr_3.4_N_7.0_), V (Fe_88.7_Zr_4.4_N_6.9_), and VI (Fe_88.7_Zr_4.4_N_6.9_) were produced by dc reactive magnetron sputtering. The mixed structure (nanocrystalline + amorphous, which, more likely, is amorphous in terms of XRD) is formed in the films. The crystalline phases, which are presented by the bcc supersaturated solid solution αFe(Zr,N) (films II, III) or bcc αFe(Zr,N) and fcc Fe_4_N (films IV, V), the two layers, the bottom amorphous and the top crystalline (films III) and the grain columnar agglomerates tending to be oriented along the film growth direction (films II, IV), are formed in the films. The annealing of the films leads to the nanocrystallization of the amorphous phase, the depletion of the bcc αFe phase of Zr and N and formation of the more pronounced columnar structure.

All films demonstrate high hardness (13.7–20.8 GPa) in the as-deposited and annealed states. The annealing leads to the decrease in the hardness, which is due to (1) a transformation of the high compressive stresses formed in the as-deposited films to tensile stresses of lower values, (2) a decrease in the effect of solid solution hardening due to the depletion of the αFe-based solid solution of Zr and N, and (3) the intensification of the columnar structure. It is shown that the values of the elastic modulus of films IV, V in the as-deposited and annealed states are lower (157–166 GPa) than that of bulk Fe (200 GPa). 

The complex structure of the films manifests itself in their magnetic properties. The hysteresis loops indicate that a significant part of the volume of each film has out-of-plane magnetic anisotropy in all as-deposited films. This can be caused by the columnar agglomeration of grains (films II and IV) or by compressive residual stresses (films I, II, and V) combined with positive magnetostriction of the ferromagnetic phase. The films III have the most complex magnetic anisotropy, which is expressed in the coupling field in the hysteresis loop (the presence of a coupling field holding part of the sample magnetization in one direction) of the as-deposited and annealed at 300 and 400 °C films.

With regard to magnetic recording, we note that industrial heads have a high saturation induction of 1.8–2.2 T due to the use of FeCo alloy [2,67,68,69], whereas our alloy free of expensive cobalt has the comparable induction in weaker magnetic fields. This realizes efficient operation with a lower amperage, a smaller cross-section of electrical conductor, and a smaller number of magnetizing winding turns, which allow one to reduce the size and weight of devices while maintaining their operational characteristics.

## Figures and Tables

**Figure 1 materials-15-00137-f001:**
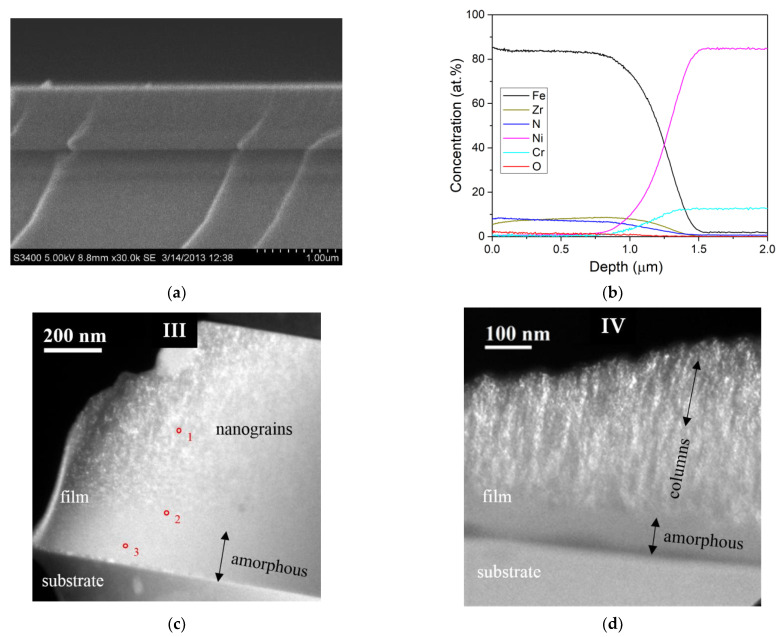
The SEM cross-sectional image (**a**) and the GDOES profile of the distribution of the elements in depth (**b**) of the as-deposited film III. The dark-field TEM images of a cross-section of the films III with an equiaxial grain orientation (**c**) and IV with a columnar structure (**d**). The red dots 1, 2, 3 in the micrograph (**c**) correspond to the EDX spectra (**e**–**g**).

**Figure 2 materials-15-00137-f002:**
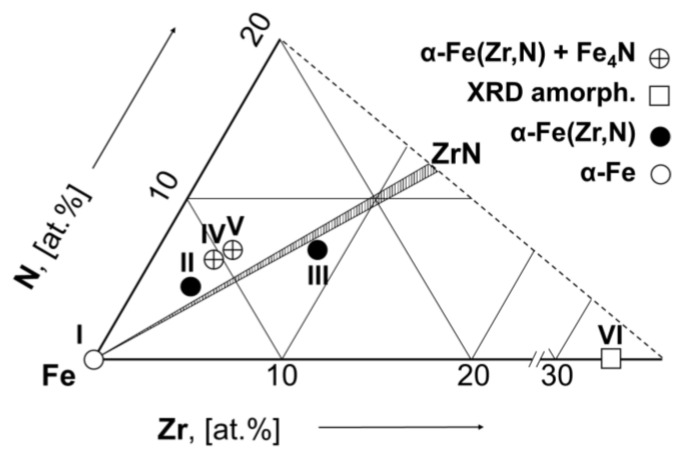
The compositions of the as-deposited films applied to the concentration triangle of the equilibrium Fe-Zr-N ternary system (scheme). The dashed area corresponds to the two-phase (αFe + ZrN) area.

**Figure 3 materials-15-00137-f003:**
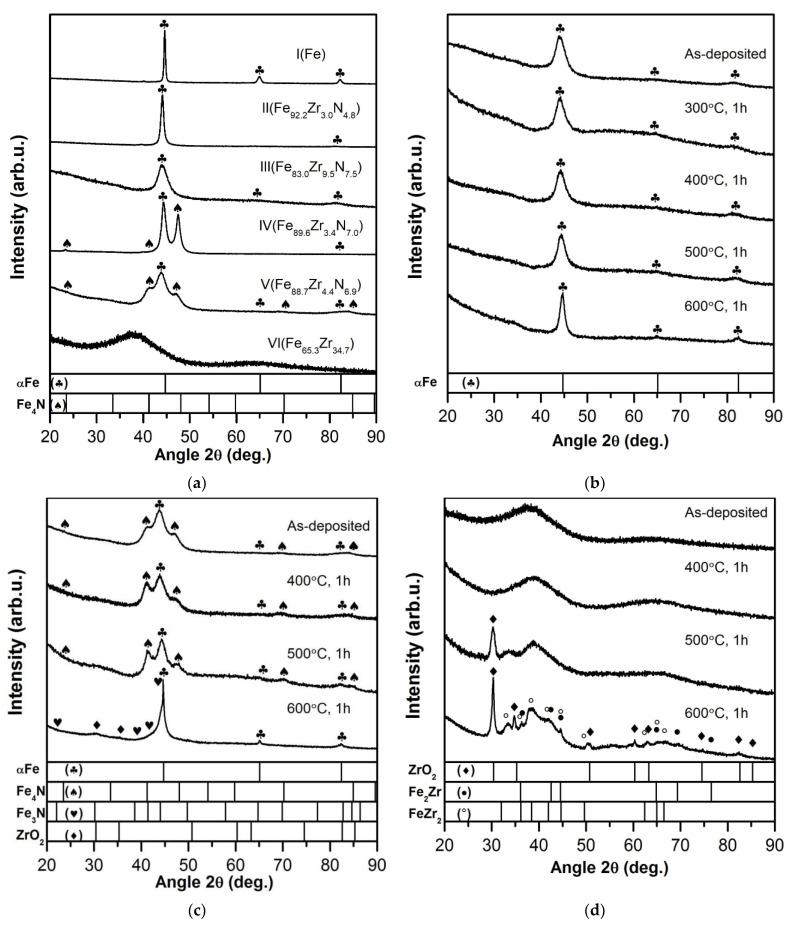
The XRD patterns of the as-deposited films (**a**) and films III (**b**), V (**c**), VI (**d**) after annealing.

**Figure 4 materials-15-00137-f004:**
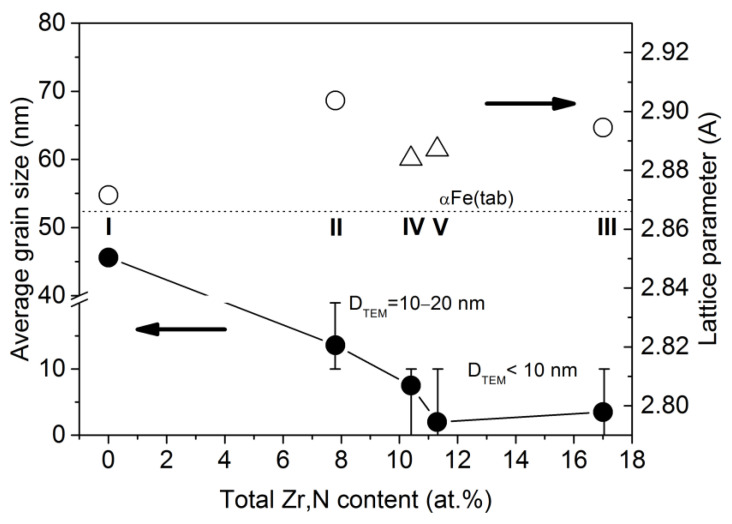
The influence of the total content of Zr and N on the lattice parameter of the bcc phase in the single-phase (open circles) and two-phase (open triangles) films and the average grain size of the bcc phase, estimated by XRD (filled circles) and TEM (hatching) in the as-deposited films.

**Figure 5 materials-15-00137-f005:**
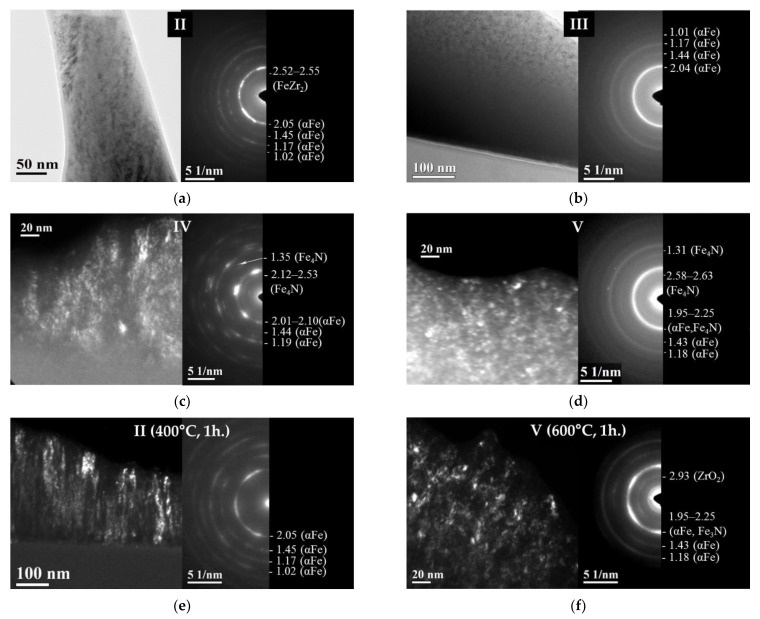
TEM images and corresponding SAED patterns taken for the as-deposited films II (**a**), III (**b**), IV (**c**), V (**d**), and films II (**e**) and V (**f**) annealed for 1 h at 400 and 600 °C, respectively. Dark-field images (**c**–**f**) were taken using the fragments of the rings, corresponding g = 110 for the bcc phase and g = 111 for the fcc phase.

**Figure 6 materials-15-00137-f006:**
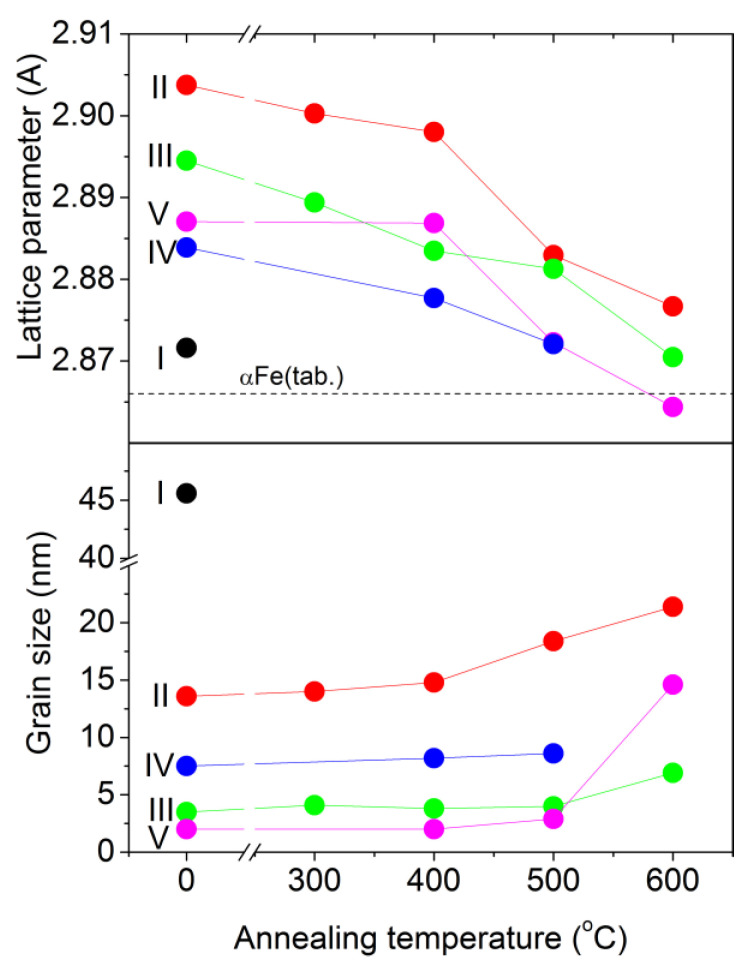
The influence of the annealing temperature on the lattice parameter and the grain size of the bcc phase.

**Figure 7 materials-15-00137-f007:**
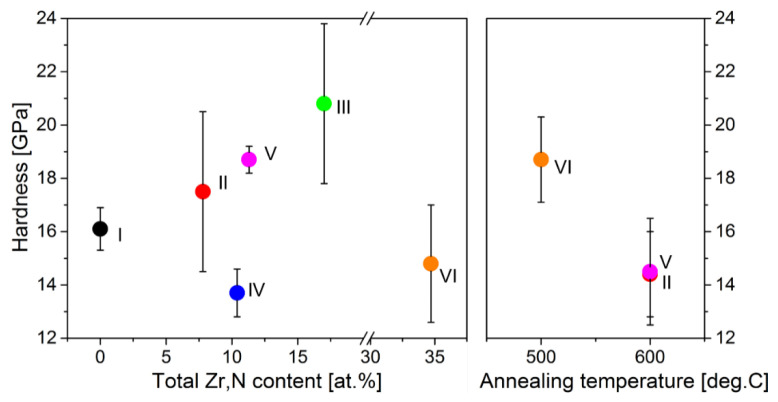
The influence of the total content of Zr and N in the films and the annealing temperature on the hardness of the films.

**Figure 8 materials-15-00137-f008:**
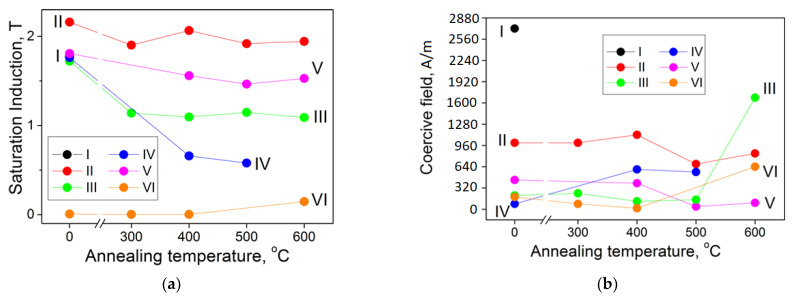
Saturation induction (**a**) and coercive field (**b**) of the films.

**Figure 9 materials-15-00137-f009:**
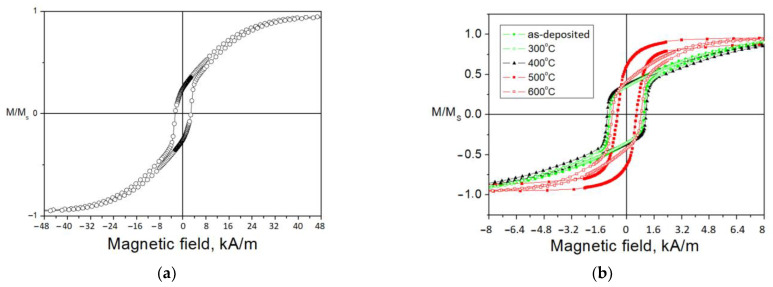
The hysteresis loops of the films I (**a**), II (**b**), III (**c**), IV (**d**), V (**e**), VI (**f**) in the as-deposited state and after annealing.

**Table 1 materials-15-00137-t001:** The results of the chemical and phase analysis of the films.

Series (*-**)	Chemical Composition	at.%N/ at.%Zr	*t*, µm	*T_a_*, °C	*v_i_* Phase	*a*, Å	*D*, nm	ε, %
I (0–0)	Fe	–	1.8	As-deposited	~1.00 α-Fe	2.872	45.6	0.268
II (2.4–5)	Fe_92.2_Zr_3.0_N_4.8_	1.6	0.9	As-deposited	~1.00 αFe(Zr,N)	2.904	13.6	0.247
				300	~1.00 αFe(Zr,N)	2.900	14.0	0.301
				400	~1.00 αFe(Zr,N)	2.898	14.8	0.270
				500	~1.00 αFe(Zr,N)	2.883	18.4	0.301
				600	~1.00 αFe(Zr,N)	2.877	21.4	0.302
III (5.3–5)	Fe_83.0_Zr_9.5_N_7.5_	0.8	0.8	As-deposited	~1.00 αFe(Zr,N)	2.895	3.5	0.172
				300	~1.00 αFe(Zr,N)	2.889	4.1	0.169
				400	~1.00 αFe(Zr,N)	2.884	3.8	0.151
				500	~1.00 αFe(Zr,N)	2.881	4.0	0.204
				600	~1.00 αFe(Zr,N)	2.871	6.9	0.075
IV (5.3–15)	Fe_89.6_Zr_3.4_N_7.0_	2.0	1.7	As-deposited	0.58 αFe(Zr,N)	2.884	7.5	0.010
					0.42 Fe_4_N	3.819	8.1	0.243
				400	0.53 αFe(Zr,N)	2.878	8.2	0.524
					0.47 Fe_4_N	3.805	8.3	0.363
				500	0.48 αFe(Zr,N)	2.872	8.6	0.432
					0.52 Fe_4_N	3.801	8.5	0.252
V (2.4–15)	Fe_88.7_Zr_4.4_N_6.9_	1.6	1.7	As-deposited	0.58 αFe(Zr,N)	2.887	2.0	0.066
					0.42 Fe_4_N	3.818	5.5	0.745
				400	0.61 αFe(Zr,N)	2.887	2.0	0.093
					0.39 Fe_4_N	3.821	6.7	0.635
				500	0.51 αFe(Zr,N)	2.872	2.9	0.131
					0.49 Fe_4_N	3.793	6.1	0.482
				600	0.28 αFe	2.864	14.6	0.038
					0.57 Fe_3_N	–	3.4	0.164
					0.15 ZrO_2_	5.087	3.7	0.379
VI (13.4–0)	Fe_65.3_Zr_34.7_	0	1.8	As-deposited	~1.00 XRD amorph.	–	–	–
				400	~1.00 XRD amorph.	–	–	–
				500	XRD amorph.	–	–	–
					fcc ZrO_2_	5.110	14.7	–
				600	XRD amorph.	–	–	–
					fcc ZrO_2_	5.092	25.9	–
					fcc FeZr_2_	12.092	5.4	–
					fcc Fe_2_Zr	7.019	16.4	–

* The Zr content in the Fe-Zr target used for deposition, wt.%; ** The N2 content in the gaseous atmosphere used for deposition, vol.%.

**Table 2 materials-15-00137-t002:** Mechanical properties of the films.

Series	*T*_a_, °C	*H*, GPa	*E*, GPa	*W*, %
I	As-deposited	16.1	174	55
II	As-deposited	17.5	292	83
	600	14.4	231	78
III	As-deposited	20.8	206	71
IV	As-deposited	13.7	157	57
V	As-deposited	18.7	166	73
	600	14.5	161	69
VI	As-deposited	14.8	249	73
	500	18.7	211	72

## Data Availability

The data presented in this study are available in this article.

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
