# Peer review of "FeZrN Films: Magnetic and Mechanical Properties Relative to the Phase-Structural State"

_materials, 2021, doi:10.3390/ma15010137_

Round 1

Reviewer 1 Report

The manuscript entitled "FeZrN films: magnetic and mechanical properties relative to the phase-structural state" concerns the analysis of pristine and annealed FeZrN films with thickness of about 1 micron. The authors utilized well established measurement techniques to investigate the structure, phase composition, as well as magnetic and mechanical properties of the films. In most cases, as-deposited films were amorphous, while annealing caused their crystallization and the formation of grains of a few nanometers in size.

FeZrN is well known material that was intensively researched 20-30 years ago. 9 papers written by the authors and self-cited in the manuscript also prove that the work deals with a topic that has already been largely exploited. Moreover, the motivation of research (magnetic tape-recording heads) is now almost lifeless. Magnetic tapes no longer play a significant role in magnetic recording technology, and publications on this subject are less and less frequent. Therefore, I think the manuscript is not scientifically sound and I expect readers' interest in the manuscript to be low.

In addition, the results contained in the study are partially a repetition of the studies contained in the publication: E. N. Sheftel et al. "Magnetic structure and magnetic properties of nanocrystalline and amorphous Fe-Zr-N films" Phys. B: Condens. Matter 2016, 494, 13-19. Therefore, I rate the novelty of the manuscript also low.

Specific comments:

  1. It is not clear how the authors planned the experiment. Both the compositions and thicknesses of the samples change in a non-obvious and non-systematic way. For example, the thickness varies from 0.8 micron to 1.8 micron with simultaneous changes in composition, which makes direct comparison of the samples impossible. Similarly, some samples were annealed to 500°C, some to 600°C, and some not at all, which is also not clear. Usually, the experiments involve changing only one independent parameter in systematic way so that it is possible to compare the samples and determine which factors influence the results.
  2. The authors claim that: "The profiles of the elemental distribution over the cross-section of the films, obtained by the GDOES method, indicate a uniform distribution of elements over the depth of the film (Figure 1b)." (page 5, line 133). Further, however, the opposite is stated: “According to the EDX data, there is an increased content of Zr in the amorphous layer, relative to the average in the film, and the gradient along the thickness of the amorphous layer from 13.5 at.% ( Figure 1c, point 3) to 11.9 (Figure 1c, point 2) and 9.8 (Figure 1c, point 1) in the direction from the substrate to the film surface ”. (page 5, line 143). The authors should explain where this discrepancy comes from. They should also conclude whether the elemental distribution is homogeneous or not.
  3. Presentation of structural results is not carried out in a systematic way. For example, it is not clear why XRD patterns are shown for only two annealed samples. It would be more clear and interesting for reader to see the evolution of the spectra with temperature, in particular for sample VI, which would allow to investigate the influence of thermal ordering on its structure. Additionally, it would be beneficial to present SAED data and XRD patterns for the same set of samples.
  4. Figure 1c duplicates Figure 1d. Figure 5a duplicates Figure 5b. Figure 5c duplicates Figure 5d.
  5. The authors correctly state that "To obtain high wear resistance, in addition to high hardness, nanocrystalline films must have a low elastic modulus E and high elastic recovery W". However, measurements showed that annealing deteriorates all these parameters. Therefore, it is not clear what the advantages of this approach are and why the authors devoted so much space in the manuscript to describe changes caused by the annealing. In the introduction, the authors also did not explain why they decided to study thermal changes of Fe-Zr-N materials.
  6. The authors claim that "The films III have the most complex magnetic anisotropy (Figure 9), which is expressed in the bias of the hysteresis loop (the presence of a bias field holding part of the sample magnetization in one direction) in the as-deposited film and after annealing at 300 and 400°C. The reason for such a bias may be the antiparallel alignment of magnetizations in alternating layers of a multilayer film [39]” (page 12, line 339).
    However, there is no visible bias on the presented hysteresis loop. Instead, a two-phase behavior can be noticed, similar to that for sample VI annealed at 600°C. The authors' interpretation is questionable, and the cited work [39] concerns other magnetic systems.
  7. The authors showed magnetic measurements for magnetically very soft samples and determined the coercivity fields with the accuracy of single Oe. Please provide information on the measurement uncertainty of the magnetic field as well as on the self remanence of the magnet. Please also describe whether the correction procedure was performed due to the magnet's remanence, and if so, how the authors did it. Additionally, experimental methods should contain information on the measurement geometry (perpendicular or parallel to the film).

Author Response

Reply to reviewer 1

The article text has undergone significant changes taking into account the changes made in response to the comments of the reviewers and the polishing of the presentation in English.

FeZrN is well known material that was intensively researched 20-30 years ago. 9 papers written by the authors and self-cited in the manuscript also prove that the work deals with a topic that has already been largely exploited. Magnetic tapes no longer play a significant role in magnetic recording technology, and publications on this subject are less and less frequent. Therefore, I think the manuscript is not scientifically sound and I expect readers' interest in the manuscript to be low”. Moreover, the motivation of research (magnetic tape-recording heads) is now almost lifeless.

 The introduction text is changed and some additional references are included to do more convincing, transparent and adequate description of the situation concerning to the Fe-Zr-N film topic and the current tape-head technology.

In addition, the results contained in the study are partially a repetition of the studies contained in the publication: E. N. Sheftel et al. "Magnetic structure and magnetic properties of nanocrystalline and amorphous Fe-Zr-N films" Phys. B: Condens. Matter 2016, 494, 13-19. Therefore, I rate the novelty of the manuscript also low.

The Fe-Zr-N films belong to the promising type of the soft magnetic Fe-MeIV-X alloy films.  Wherein the Fe-Zr-N films are the fruitful object to research relations between the deposition conditions, temperature of subsequent annealing, the chemical and phase compositions of films, their structure and properties.  Therefore, the development of the research presented in the article published in Physica B: Condens. Matter as we think to be rational.

Novelty of the presented data, as compared to the data published in Physica B: Condens. Matter, consists in the following:1) The films were prepared in the other sputtering-deposition processes wherein the target construction and composition as well as the N2/Ar proportion in the gas mixture were not changed and the chamber was prepared for obtaining the purer films in terms of the oxygen impurity content.

2) The lattice parameters of the phases were calculated based on the positions of the gravity centers of the high-angle reflections using the Voigt approximation. This made it possible to significantly reduce the error of determination of the lattice parameter.

3) The structure is studied by Transmission electron microscopy (TEM) and electron diffraction (ED) methods and based on the analysis of the behavior of hysteresis loops as well.4) For the first time in the published literature, the data on the mechanical properties (hardness H, elastic modulus E, and elastic recovery W) of the FeZrN film are presented and the interconnection of the parameters with the phase composition and magnetic properties of the films is discussed. 

Specific comments:

  1. It is not clear how the authors planned the experiment. Both the compositions and thicknesses of the samples change in a non-obvious and non-systematic way. For example, the thickness varies from 0.8 micron to 1.8 micron with simultaneous changes in composition, which makes direct comparison of the samples impossible.

The deposition duration was controlled under the deposition experiments (it was added to Section 2). Wherein the deposition rate and the film thickness vary depending on the cathode chemical composition and the gas atmosphere. The deposition duration control is applied quite often in studies including the deposition process.

Similarly, some samples were annealed to 500°C, some to 600°C, and some not at all, which is also not clear. Usually, the experiments involve changing only one independent parameter in systematic way so that it is possible to compare the samples and determine which factors influence the results.

The following sentences were added to the section 2:

The films as-sputtered and annealed at 400, 500 and 600°C (excluding films IV that were annealed at 400 and 500°C) were studied. In terms of the present work, the choice of the annealing temperatures equal to 400, 500 and 600°C is based on the results of our previous works [32-34], in which it was shown that the magnetic properties (Bs and Hc) of the as-deposited Fe-Zr-N films change during annealing at the above temperatures. The annealing at 400 or 500°C (in accordance with the film composition) provides the best combination of the film soft magnetic properties that begin to get worse at 600°C due to the diffusion interaction of the film with the substrate. As for the films I(Fe), they were obtained only in as-deposited state for the study of the influence of Zr and N additions on structure and properties of Fe films. It is obvious that annealing of as-deposited Fe films does not improve their properties.

  1. The authors claim that: "The profiles of the elemental distribution over the cross-section of the films, obtained by the GDOES method, indicate a uniform distribution of elements over the depth of the film (Figure 1b)." (page 5, line 133). Further, however, the opposite is stated: “According to the EDX data, there is an increased content of Zr in the amorphous layer, relative to the average in the film, and the gradient along the thickness of the amorphous layer from 13.5 at.% ( Figure 1c, point 3) to 11.9 (Figure 1c, point 2) and 9.8 (Figure 1c, point 1) in the direction from the substrate to the film surface ”. (page 5, line 143). The authors should explain where this discrepancy comes from. They should also conclude whether the elemental distribution is homogeneous or not.

The end of the Section 3.1 was expanded with the following:

Note that it is impossible to see the features of the chemical composition in the amorphous layer with a thickness no more than 200 nm in the GDOES curves (Figure 1b), since it is spread over the entire depth (more than 0.5 μm) of the roughness of the Ni-Cr substrate.

  1. Presentation of structural results is not carried out in a systematic way. For example, it is not clear why XRD patterns are shown for only two annealed samples. It would be more clear and interesting for reader to see the evolution of the spectra with temperature, in particular for sample VI, which would allow to investigate the influence of thermal ordering on its structure. Additionally, it would be beneficial to present SAED data and XRD patterns for the same set of samples.

The XRD patterns of the annealed films are added to Figure 3.

The results of the disruptive and relatively laborious TEM study are in good agreement with the results of the simpler XRD method. Moreover, the TEM data rather complement and confirm the XRD data; therefore, the authors believe that the TEM data presented in this work should be sufficient to describe the structure of the films.

Films in the initial state, for which the TEM studies were carried out, were selected as of interest in terms of functional properties (proximity to the two-phase Fe+ZrN area in equilibrium ternary phase diagram Fe-Zr-N, Figure 2, allowing to predict high Bs and low Hc). For the films in the annealed state, the focus was made on the temperature range 400-600°C, which, according to the data of previous studies, provides the best complex of the soft magnetic properties that begin to get worse at 600°C due to diffusion interaction of the film and the substrate. That is why the relation between the phase composition, structure and properties of the as-deposited films and annealed at 400, 500 and 600°C were studied in the present work.

  1. Figure 1c duplicates Figure 1d. Figure 5a duplicates Figure 5b. Figure 5c duplicates Figure 5d.

Sorry for our mistake. The proper figures are inserted.

  1. The authors correctly state that "To obtain high wear resistance, in addition to high hardness, nanocrystalline films must have a low elastic modulus E and high elastic recovery W". However, measurements showed that annealing deteriorates all these parameters. Therefore, it is not clear what the advantages of this approach are and why the authors devoted so much space in the manuscript to describe changes caused by the annealing. In the introduction, the authors also did not explain why they decided to study thermal changes of Fe-Zr-N materials.

Annealing of the films actually leads to a decrease in the hardness and elastic recovery, although these parameters remain sufficiently high for soft magnetic films. The most important, from the point of view of the effect on the magnetic properties, is the low elastic modulus, which becomes noticeably lower as a result of annealing.

  1. The authors claim that "The films III have the most complex magnetic anisotropy (Figure 9), which is expressed in the bias of the hysteresis loop (the presence of a bias field holding part of the sample magnetization in one direction) in the as-deposited film and after annealing at 300 and 400°C. The reason for such a bias may be the antiparallel alignment of magnetizations in alternating layers of a multilayer film [39]” (page 12, line 339).

However, there is no visible bias on the presented hysteresis loop. Instead, a two-phase behavior can be noticed, similar to that for sample VI annealed at 600°C. The authors' interpretation is questionable, and the cited work [39] concerns other magnetic systems.

The following two sentences were added to the paragraph quoted by the reviewer:

Note that the film VI annealed at 600°C has a relative remanence of 0.64, whereas the film III annealed at 400°C has that of 0.07. However, despite such a low remanence, sample III annealed at 400°C does not show perpendicular anisotropy, and at the same time, the magnetization and demagnetization curves near the coupling field run almost vertically.

In addition, in Figure 9c the horizontal scale is doubled to make the details of the hysteresis better visible.

Ref. [39] is replaced by Ch. Wang, Sh. Zhang, Y. Huang, T. Sang, D. Cao, X. Wang, J. Xu, G. Zhao, C. Wang, Sh. Li, Dynamic interlayer exchange coupling in magnetic anisotropic FeCoB/Ru/FeCoB sandwich trilayers, Journal of Magnetism and Magnetic Materials, Volume 527, 2021, 167801, https://doi.org/10.1016/j.jmmm.2021.167801.

  1. The authors showed magnetic measurements for magnetically very soft samples and determined the coercivity fields with the accuracy of single Oe. Please provide information on the measurement uncertainty of the magnetic field as well as on the self remanence of the magnet. Please also describe whether the correction procedure was performed due to the magnet's remanence, and if so, how the authors did it. Additionally, experimental methods should contain information on the measurement geometry (perpendicular or parallel to the film).

Added to the end of Materials and methods:

The vibrating-sample magnetometer uses a Hall sensor to measure the magnetic field in the gap of an electromagnet regardless of the current strength and remanence of the electromagnet. The hysteresis loops are built according to the Hall sensor readings, and the current in the electromagnet is used only to correlate the Hall sensor readings with the measured magnetic moment. Note that all measurements started with a maximum field of 1280 kA/m, then, after passing through the zero field, a negative field of 1280 kA/m was reached and the cycle was closed in reverse order. The accuracy of the magnetic field measuring was no worse than 10 A/m. All measurements were made in the film plane.

Reviewer 2 Report

The authors present an interesting work that highlight the correlation between the magnitudes of the magnetic and mechanical properties of FeZrN films of various compositions and crystalline structures, obtained by dc reactive magnetron sputtering. Before publication, there are some aspects that deserve more attention:

  1. The authors use simultaneously SI and CGS Units. The use of SI Units is one of the rules of Materials and should be followed. Please consider changing from Oe to A/m. Also σ = 1010 erg/cm3 (line 324) should be in Pa;
  2. EDX (data not shown). “According to the EDX data, there is an increased content of Zr in the amorphous layer, relative to the average in the film, and the gradient along the thickness of the amorphous layer from 13.5 at.% (Figure 1c, point 3) to 11.9 (Figure 1c, point 2) and 9.8 (Figure 1c, point 1) in the direction from the substrate to the film surface. This is consistent with the observed increase in the thickness of the amorphous layer in the films with a high Zr content.” This statement should be better explained since no EDX is presented and is not possible to visualize the mentioned gradient changes.
  3. There is no mention about the possible influence of the elements Ni and Cr evidenced in b)-Figure-1 (diffused from the substrate into the film) on the evaluated properties;
  4. It is important identify in Table 1 the films who were deposited in glass substrate from that deposited in Ni-Cr alloy;
  5. Line 128: “All films contain impurity oxygen in an amount not exceeding 2 at.%.”. It is not clear about the oxygen (at%) could be considered an impurity since ZrO2 is found in the films composition. EDX spectra, if presented, could also be used to measure the amounts of elements present.

Author Response

Reply to reviewer 2

The article text has undergone significant changes taking into account the changes made in response to the comments of the reviewers and the polishing of the presentation in English.

  1. The authors use simultaneously SI and CGS Units. The use of SI Units is one of the rules of Materials and should be followed. Please consider changing from Oe to A/m. Also σ = 1010 erg/cm3 (line 324) should be in Pa;

Corresponding corrections were made in the text. In Figures 8 and 9, the CGS units are replaced by SI.

The magnetic values are changed in the abstract and the following sentences:

The magnetoelastic anisotropy field 3λsσ/Ms (Ms = 1.3 T, λs = 10-5 is the saturation magnetostriction, and σ = 109 Pa is the residual stresses [31]) does not exceed 24 kA/m for the films, and the perpendicular anisotropy in all films, except for films I and VI, fits into this range.

The topological coupling field is analytically expressed in the "orange peel" model [43,44] and is equal to Hcf = (21/2λtfe)-1π2l2Msexp(-23/2πtd/λ) [64], where l and λ are amplitude and wavelength of roughness at the interface, respectively (in our case, they are equal to the grain size [65], Table 1); Ms is saturation magnetization (1.8 T); and tfe and td are the thicknesses of the ferromagnetic layer and the non-ferromagnetic layer (the “dead” layer on the surface of the ferromagnetic layer does not exceed 1 nm [66]), respectively. After substitution of values suitable for the films III, we obtain Hcf equal to ~240 A/m, which is close to the observed value of the coupling field (Figure 9).

  1. EDX (data not shown). “According to the EDX data, there is an increased content of Zr in the amorphous layer, relative to the average in the film, and the gradient along the thickness of the amorphous layer from 13.5 at.% (Figure 1c, point 3) to 11.9 (Figure 1c, point 2) and 9.8 (Figure 1c, point 1) in the direction from the substrate to the film surface. This is consistent with the observed increase in the thickness of the amorphous layer in the films with a high Zr content.” This statement should be better explained since no EDX is presented and is not possible to visualize the mentioned gradient changes.

The EDX spectra are added to Figure 1 and the corresponding references to Figure 1 are added to Section 3.1.

  1. There is no mention about the possible influence of the elements Ni and Cr evidenced in b)-Figure-1 (diffused from the substrate into the film) on the evaluated properties;

The following sentences are added to Section 2:

The Ni-Cr substrates were used exclusively for GDOES measurements on the as-deposited films. The rest of the measurements were performed on the glass substrates.

  1. It is important identify in Table 1 the films who were deposited in glass substrate from that deposited in Ni-Cr alloy;

We think our answer to the previous comment makes this clear.

  1. Line 128: “All films contain impurity oxygen in an amount not exceeding 2 at.%.”. It is not clear about the oxygen (at%) could be considered an impurity since ZrO2 is found in the films composition. EDX spectra, if presented could also be used to measure the amounts of elements present.

The following sentences are added to 2nd paragraph of Section 3.2.2:

Oxygen, characterized by a high affinity for Zr (one of the highest formation energies of -355 kJ/mol∙atom), with its corresponding impurity content, should form an oxide phase both in the as-deposited state and in the annealed state at temperatures below 500°C. The appearance of the ZrO2 phase only in the films annealed at temperatures above 500°C (Table 1) indicates that the formation of this phase is not the result of the influence of impurity oxygen.

Reviewer 3 Report

The authors have performed an extended study on a batch of FeZrN samples with the aim of developing/improving materials for writing head materials for magnetic tape recording. The characterisation fabrication and characterisation techniques used are adequate for the study. The work could be interesting for people working in the field of magnetic materials for these applications. However, the figures and figure captions are not well explained and several times, figure panels are repeated and not showing what is mentioned in the text of the manuscript. This makes it very difficult to follow and also confusing. This is an error that must be resolved. On the other hand, although the structural analysis is quite thorough, there are important points in the discussions missing and, under my point of view, lack of important information in several parts of the work.
Due to this I recommend to reconsider the manuscript after MAJOR REVISION.
Here I indicate several points that should be addressed prior resubmission.

1) The introduction must be improved by indicating how this work fits in the special issue addressed of Magnetic Materials for Power Electronic Applications, as magnetic recording, and specially tapes are focused in specifically saving energy and are not associated to Power Electronics.

2) In Figure 1 panels c) and d) are the same. This creates a lot of confusion and is not supporting what is written in the text.
3) Also, in figure 1 it is interesting to see how sharp the interface between the substrate and the film is in panel a) and how wide this interface appears in the GDOES profile. Could the authors comment about this? What is the real in-thickness resolution of GDOES? Could it be possible to deconvolve the plasma milling probe profile and improve the accuracy of the profile measurement?
4) It is indicated in the manuscript that the GDOES is used for the detection of light elements such as N and O. O is missing in the profile analysis, and it is mentioned later in the manuscript to form ZrO. Have the authors been able to detect the O impurities by using GDOES to support this point? In the text is also indicated that the O impurities does not exceed a 2%, but this is not supported with any experimental data up to my understanding of the manuscript.
5) It is very important at this point also to state clearly which samples are being compared because the reader really do not know it till section 3.2 where annealed and as deposited are separated in two sub-sections. What are the conditions of the samples analysed in figures 1 and 2?
6) The black lines in the composition triangle of figure 2 are not described in the caption and in the text. Are those representing the stable phase the authors mention in the main text? Please clarify.
7) There are several characters in section 3.2.1 which I do not understand in between the angles indicating the diffraction peak positions. This is very minor.
8) Have the authors considered to represent the Intensity axis of the XRDs in figure 4 in logarithmic scale? Otherwise, it is almost impossible to observe the peaks they have identified in the diffractograms.
9) In figure 5 panels a and b and c and d are the same and are repeated. This must be corrected as it makes again a lot of confusion to follow the manuscript and the explanations.
10) In figure 5, it should be good to indicate which peak of the ED is being used to record the DF image and indicate to which structure it corresponds. Also, indicate in which region of the sample are the SAED patterns recorded for clarity.
11) In the magnetic properties analysis, IS and CGS units are being mixed. I have no problem with CGS or IS, but for coherence, it would be good to be consistent and use either one or the other.
12) It would be good to know the field application orientation during the VSM measurements, as the authors are commenting on the magnetic anisotropy of their samples. Have the authors performed a study varying the relative orientation between the field and the sample? Are the presented loops corresponding to in-plane field application? For the discussion about perpendicular magnetic anisotropy, it would be good to have orthogonal measurements of the loops with in-plane field, and an out-of-plane field one to completely support the PMA presence in the system.
13) In figure 9 c) it is mentioned that this one is the most interesting in terms of anisotropy. I agree the loop is interesting, but it is unprecise in my opinion to say that the loop has bias. I understand what the authors are trying to say, but this can bring confusion as having a bias in the hysteresis loop is not what the authors have in their measurement. They have a SAF-like hysteresis loop which has no bias. It is like in remanence they have some sort of AFM coupling between different parts of the system which are compensating each other.
14) Finally, I would like to ask the authors to make a comparison of the properties of their material in the framework of actual materials used for writing heads for tape recording, thus this can bring context to the work and state where the discussed system is.

Author Response

Reply to reviewer 3

The article text has undergone significant changes taking into account the changes made in response to the comments of the reviewers and the polishing of the presentation in English.

1) The introduction must be improved by indicating how this work fits in the special issue addressed of Magnetic Materials for Power Electronic Applications, as magnetic recording, and specially tapes are focused in specifically saving energy and are not associated to Power Electronics.

The introduction is expanded with the following:

Note that the use of magnetic materials for power electronic applications, in particular in inductors and transformers [9], requires the same properties of magnetic materials, namely, the low coercive field, high saturation induction, and high resistivity for high magnetic permeability in rf magnetic fields [10-13], which allow the size of the converter, while maintaining the high current, to be reduced [14].

2) In Figure 1 panels c) and d) are the same. This creates a lot of confusion and is not supporting what is written in the text.

Sorry for our mistake. The proper figures are inserted.

3) Also, in figure 1 it is interesting to see how sharp the interface between the substrate and the film is in panel a) and how wide this interface appears in the GDOES profile. Could the authors comment about this? What is the real in-thickness resolution of GDOES? Could it be possible to deconvolve the plasma milling probe profile and improve the accuracy of the profile measurement?

As added in the second paragraph of section 2, The Ni-Cr substrates were used exclusively for GDOES measurements on the as-deposited films.

The GDOES distribution curves are not suitable for measuring the film thickness, studying the film-substrate interface and layers near this interface, since the Ni-Cr alloy is characterized by a significantly higher roughness (about 0.5 μm, Figure 1b) as compared to that of the glass substrate. The film thickness was estimated from the SEM cross-sectional image (Figure 1a), as mentioned in Section 2.

4) It is indicated in the manuscript that the GDOES is used for the detection of light elements such as N and O. O is missing in the profile analysis, and it is mentioned later in the manuscript to form ZrO. Have the authors been able to detect the O impurities by using GDOES to support this point? In the text is also indicated that the O impurities does not exceed a 2%, but this is not supported with any experimental data up to my understanding of the manuscript.

The GDOES profile of the oxygen depth distribution is added in Figure 1b. As mentioned in Section 3.2.2, the formation of the fcc ZrO2 phase in films V, VI could be related to both the presence of impurity oxygen in the films and its diffusion from the substrate to the film at 500, 600°C [43]. Oxygen, characterized by a high affinity for Zr (one of the highest formation energies of -355 kJ/mol∙atom), with its corresponding impurity content, should have formed an oxide phase both in the as-deposited state and in the annealed state at temperatures below 500°C. However, the appearance of the ZrO2 phase only in the films annealed at temperatures above 500°C (Table 1) indicates that the formation of this phase is not the result of the influence of impurity oxygen.

5) It is very important at this point also to state clearly which samples are being compared because the reader really do not know it till section 3.2 where annealed and as deposited are separated in two sub-sections. What are the conditions of the samples analysed in figures 1 and 2?

Corresponding corrections were made both in the text and the figure captions. "as-deposited films" has been added in sections 2 and 3.2 and in the caption to Figures 1 and 2.

6) The black lines in the composition triangle of figure 2 are not described in the caption and in the text. Are those representing the stable phase the authors mention in the main text? Please clarify.

The following sentence is added to Figure 2 caption: “The dashed area corresponds to the two-phase (αFe + ZrN) area.”

7) There are several characters in section 3.2.1 which I do not understand in between the angles indicating the diffraction peak positions. This is very minor.

The typo has been fixed. The character º in section 3.2.1 is replaced by °.

8) Have the authors considered to represent the Intensity axis of the XRDs in figure 4 in logarithmic scale? Otherwise, it is almost impossible to observe the peaks they have identified in the diffractograms.

Unfortunately, the use of a logarithmic axis instead of a linear one does not simplify the perception of the XRD peaks.

Linear scale of intensity

Logarithmic scale of intensity

9) In figure 5 panels a and b and c and d are the same and are repeated. This must be corrected as it makes again a lot of confusion to follow the manuscript and the explanations.

Sorry for our mistake. The proper figures are inserted.

10) In figure 5, it should be good to indicate which peak of the ED is being used to record the DF image and indicate to which structure it corresponds. Also, indicate in which region of the sample are the SAED patterns recorded for clarity.

The Figure 5 caption is changed:

TEM images and corresponding SAED patterns obtained for the as-deposited films II (a), III (b), IV (c), V (d) and films II (e) and V (f) annealed for 1 hour at 400 and 600°C respectively. Dark-field images (c-f) were taken using the fragments of the rings, corresponding g=110 for the bcc phase and g=111 for the fcc phase.

11) In the magnetic properties analysis, IS and CGS units are being mixed. I have no problem with CGS or IS, but for coherence, it would be good to be consistent and use either one or the other.

CGS units are changed to SI in the abstract, Section 2 and Section 3.4.

12) It would be good to know the field application orientation during the VSM measurements, as the authors are commenting on the magnetic anisotropy of their samples. Have the authors performed a study varying the relative orientation between the field and the sample? Are the presented loops corresponding to in-plane field application? For the discussion about perpendicular magnetic anisotropy, it would be good to have orthogonal measurements of the loops with in-plane field, and an out-of-plane field one to completely support the PMA presence in the system.

The end of Section 2 is extended and the following two sentences are added to Section 3.4:

Measurements of hysteresis loops in different directions in the film plane and perpendicular to the film plane are not shown in order to Figure 9 was not cluttered up. Measurements of hysteresis loops in different directions in the film plane are in good agreement with each other. The perpendicular anisotropy did not exceed several hundred oersteds; therefore, it is very difficult to analyze it on the hysteresis loops measured perpendicular to the film plane on the background of the shape anisotropy field 4πMs.

13) In figure 9 c) it is mentioned that this one is the most interesting in terms of anisotropy. I agree the loop is interesting, but it is unprecise in my opinion to say that the loop has bias. I understand what the authors are trying to say, but this can bring confusion as having a bias in the hysteresis loop is not what the authors have in their measurement. They have a SAF-like hysteresis loop which has no bias. It is like in remanence they have some sort of AFM coupling between different parts of the system which are compensating each other.

The term “bias field” is replaced by a “coupling field”.

14) Finally, I would like to ask the authors to make a comparison of the properties of their material in the framework of actual materials used for writing heads for tape recording, thus this can bring context to the work and state where the discussed system is.

The conclusions is expanded with the following:

With regard to magnetic recording, we note that industrial heads have a high saturation induction of 1.8-2.2 T due to the use of FeCo alloy [2,67-69], whereas our alloy free of expensive cobalt has the comparable induction in weaker magnetic fields. This realizes efficient operation with lower amperage, a smaller cross-section of electrical conductor, a smaller number of magnetizing winding turns, which allow one to reduce the size and weight of devices while maintaining their operational characteristics.

Round 2

Reviewer 1 Report

Thank you for addressing in detail all the questions and comments raised in the review. The answers are fully satisfactory and corrections introduced by the authors increased the quality of the manuscript. I belief, however, a few minor issues concerning magnetism should be addressed before the paper is accepted for publication:

  1. Page 13, line 372. “Measurements of hysteresis loops in different directions in the film plane and perpendicular to the film plane are not shown in order to Figure 9 was not cluttered up.” Measurements in the direction perpendicular to the film plane are very important in the context of the magnetic anisotropy discussion. Therefore, I strongly encourage the authors to show such results. I believe the authors will find a way to present them in a clear form.
  2. Page 13, line 376. “The perpendicular anisotropy did not exceed several hundred oersteds (…)” Do the authors mean the anisotropy field here? If so, please correct.
  3. In paragraph 3.4, the authors mentioned several times perpendicular magnetic anisotropy. However, one cannot speak about a perpendicular anisotropy without presenting the hysteresis loops in a perpendicular direction supported by comparing them with loops in a direction parallel to the sample surface. The current claims about perpendicular anisotropy are therefore only assumptions and are not evidence based. Therefore, I suggest rephrasing the relevant sentences in paragraph 3.4 or add the mentioned hysteresis loops (see the point No. 1).

Author Response

Reply to reviewer 1

Thank you for addressing in detail all the questions and comments raised in the review. The answers are fully satisfactory and corrections introduced by the authors increased the quality of the manuscript. I belief, however, a few minor issues concerning magnetism should be addressed before the paper is accepted for publication:

  1. Page 13, line 372. “Measurements of hysteresis loops in different directions in the film plane and perpendicular to the film plane are not shown in order to Figure 9 was not cluttered up.” Measurements in the direction perpendicular to the film plane are very important in the context of the magnetic anisotropy discussion. Therefore, I strongly encourage the authors to show such results. I believe the authors will find a way to present them in a clear form.

First and foremost, we would like to thank the Reviewers for their insightful comments. These comments, indeed, improved the paper content and presentation considerably, and we feel indebted to the Reviewers for the time spent for thorough and constructive review of the manuscript. Please find below the detailed responses to every comment made by each of the Reviewers. We sincerely hope that these detailed responses will fully satisfy the Reviewers as well as the Editor.

Taking into account all the comments and suggestions regarding perpendicular anisotropy noted by the respected reviewers 1 and 3, we decided to exclude the mentions of perpendicular anisotropy and to  use terms “the anisotropy field” (do not to discuss its symmetry), where necessary. Changes have been made in  Section 3.4:

Complex symmetry and volumetric distribution of magnetic anisotropy in heterophase films manifest themselves in the shape of hysteresis loops (Figure 9). Measurements of hysteresis loops in different directions in the film plane are in good agreement with each other (not shown in order to Figure 9 was not cluttered up).

In all films in the as-deposited state, hysteresis loops indicate that a significant part of the volume of each film has strong anisotropy field. This can be caused by the columnar agglomeration of grains (the films II and IV) or by compressive residual stresses (the films I, II, and V) combined with positive magnetostriction of the ferromagnetic phase [56]. The magnetoelastic anisotropy field 3λsσ/Ms (Ms = 1.3 T, λs = 10-5 is the saturation magnetostriction, and σ = 109 Pa is the residual stresses [31]) does not exceed 24 kA/m for the films, and the anisotropy field in all films, except for films I and VI, fits into this range. After annealing, for all films, except for films III and IV, the coercive field decreases and the anisotropy field weakens. This indicates the predominance of the exchange interaction over the local magnetic anisotropy [27]. Note that residual stresses are not the only source of magnetoelastic anisotropy. The magnetostriction itself causes a stress Eλs (E is the elastic modulus) of ~1 MPa, which cannot be detected against the background of residual stresses. In addition, the competition between the natural magnetoelastic anisotropy (3/2)λs2E (independent of the presence of stresses) and the effective magnetic anisotropy of the material (includes all possible types of magnetic anisotropy) causes the dependence of the elastic modulus on the magnetization of the material (ΔE-effect [57]), therefore, to reduce the ΔE-effect, it is necessary to reduce the elastic modulus.

The films III have the most complex magnetic anisotropy (Figure 9), which is expressed in the coupling field in the hysteresis loop (the presence of a coupling field holding part of the sample magnetization in one direction) in the as-deposited film and after annealing at 300 and 400°C. Note that the film VI annealed at 600°C has a relative remanence of 0.64, whereas the film III annealed at 400°C has that of 0.07. However, despite such a low remanence, sample III annealed at 400°C does not show strong anisotropy field, and at the same time, the magnetization and demagnetization curves near the coupling field run almost vertically.

The reason for such a coupling may be the antiparallel alignment of magnetizations in alternating layers of a multilayer film [58], and the effect manifests itself both in the case of magnetization in the film plane [59,60], and during magnetization perpendicular to the plane [61]. Note that this effect manifests itself only in the films III, for which the amorphous layer thickness is highest (about 200 nm) and the entire film thickness is lowest (0.8 μm).

The topological coupling field is analytically expressed in the "orange peel" model [62,63] and is equal to Hcf = (21/2λtfe)-1π2l2Msexp(-23/2πtd/λ) [64], where l and λ are amplitude and wavelength of roughness at the interface, respectively (in our case, they are equal to the grain size [65], Table 1); Ms is saturation magnetization (1.8 T); and tfe and td are the thicknesses of the ferromagnetic layer and the non-ferromagnetic layer (the “dead” layer on the surface of the ferromagnetic layer does not exceed 1 nm [66]), respectively. After substitution of values suitable for the films III, we obtain Hcf equal to ~240 A/m, which is close to the observed value of the coupling field (Figure 9).

After annealing at 500 and 600°C, the amorphous layer crystallizes; this leads to an increase in the coercive field, and the anisotropy field increases due to the strengthening of the columnar structure (Section 3.2). The films IV and V also contain an amorphous layer, however, no coupling field is noticeable in their hysteresis loops, probably owing to the several times smaller thickness of the amorphous layer relative to the entire film thickness (Table 1).

  1. Page 13, line 376. “The perpendicular anisotropy did not exceed several hundred oersteds (…)” Do the authors mean the anisotropy field here? If so, please correct.

The sentence “The perpendicular anisotropy did not exceed several hundred oersteds (...)” is removed from the text, like all the others concerning the measurements of loops perpendicular to the film plane.

  1. In paragraph 3.4, the authors mentioned several times perpendicular magnetic anisotropy. However, one cannot speak about a perpendicular anisotropy without presenting the hysteresis loops in a perpendicular direction supported by comparing them with loops in a direction parallel to the sample surface. The current claims about perpendicular anisotropy are therefore only assumptions and are not evidence based. Therefore, I suggest rephrasing the relevant sentences in paragraph 3.4 or add the mentioned hysteresis loops (see the point No. 1).

Based on your suggestion and as pointed out in the answer to your 1st comment, we decided to exclude the mentions of perpendicular anisotropy and to use terms “the anisotropy field” (do not to discuss its symmetry), where necessary. Note that perpendicular loops were measured for all the films; however, they are looking about the same and the same ugly. We bring a couple of loops:

Reviewer 2 Report

The authors have addressed all the comments and suggestions I made in the first review. The quality of the article has significantly improved. I don’t have further suggestions. The paper is now acceptable for publication.

Author Response

Reply to reviewer 2

First and foremost, we would like to thank the Reviewers for their insightful comments. These comments, indeed, improved the paper content and presentation considerably, and we feel indebted to the Reviewers for the time spent for thorough and constructive review of the manuscript.

Reviewer 3 Report

The authors have addressed almost all the points raised in the previous round but there is an important point about the hysteresis loops that must be clarified. Due to this and in the view that the uthors state that the measurements are already done, I recommend Minor Changes before publication.

My comment is as follows:

1) The authors state that it is complicated to present the hysteresis loops of the system in a clear manner, specifically when discussing about the out-of-plane ones. Just an statement saying that the anisotropy field is too low to be clearly observed or measured is not enough and the authors should support their responses with data. At least sending the loops to the reviewers to seel their claims.
I recommend the authors to include the out-of-plane hysteresis loops in the manuscript to support their discussion.

I have also some confussion after reading the rest of the reports, specifically when talking again about the histeresys loops. One of the referees is asking if the measurements where performed in-plane only and I understood from their response that the answer was yes, however I asked something similar and my response was that in-plane orthogonal and out-of-plane measurements where performed. Please clarify this point.

Author Response

Reply to reviewer 3

The authors have addressed almost all the points raised in the previous round but there is an important point about the hysteresis loops that must be clarified. Due to this and in the view that the authors state that the measurements are already done, I recommend Minor Changes before publication.

My comment is as follows:

1) The authors state that it is complicated to present the hysteresis loops of the system in a clear manner, specifically when discussing about the out-of-plane ones. Just a statement saying that the anisotropy field is too low to be clearly observed or measured is not enough and the authors should support their responses with data. At least sending the loops to the reviewers to see their claims.

I recommend the authors to include the out-of-plane hysteresis loops in the manuscript to support their discussion.

I have also some confusion after reading the rest of the reports, specifically when talking again about the hysteresis loops. One of the referees is asking if the measurements where performed in-plane only and I understood from their response that the answer was yes, however I asked something similar and my response was that in-plane orthogonal and out-of-plane measurements were performed. Please clarify this point.

First and foremost, we would like to thank the Reviewers for their insightful comments. These comments, indeed, improved the paper content and presentation considerably, and we feel indebted to the Reviewers for the time spent for thorough and constructive review of the manuscript. Please find below the detailed responses to every comment made by each of the Reviewers. We sincerely hope that these detailed responses will fully satisfy the Reviewers as well as the Editor.

Indeed, the sentence “All measurements were made in the film plane” was added in Section 2 of the first round of peer review.

Taking into account all the comments and suggestions regarding perpendicular anisotropy noted by the respected reviewers 1 and 3, we decided to exclude the mentions of perpendicular anisotropy and to use terms  “the anisotropy field” ( do not to discuss its symmetry), where necessary.

Changes have been made in the Section 3.4:

Complex symmetry and volumetric distribution of magnetic anisotropy in heterophase films manifest themselves in the shape of hysteresis loops (Figure 9). Measurements of hysteresis loops in different directions in the film plane are in good agreement with each other (not shown in order to Figure 9 was not cluttered up).

In all films in the as-deposited state, hysteresis loops indicate that a significant part of the volume of each film has strong anisotropy field. This can be caused by the columnar agglomeration of grains (the films II and IV) or by compressive residual stresses (the films I, II, and V) combined with positive magnetostriction of the ferromagnetic phase [56]. The magnetoelastic anisotropy field 3λsσ/Ms (Ms = 1.3 T, λs = 10-5 is the saturation magnetostriction, and σ = 109 Pa is the residual stresses [31]) does not exceed 24 kA/m for the films, and the anisotropy field in all films, except for films I and VI, fits into this range. After annealing, for all films, except for films III and IV, the coercive field decreases and the anisotropy field weakens. This indicates the predominance of the exchange interaction over the local magnetic anisotropy [27]. Note that residual stresses are not the only source of magnetoelastic anisotropy. The magnetostriction itself causes a stress Eλs (E is the elastic modulus) of ~1 MPa, which cannot be detected against the background of residual stresses. In addition, the competition between the natural magnetoelastic anisotropy (3/2)λs2E (independent of the presence of stresses) and the effective magnetic anisotropy of the material (includes all possible types of magnetic anisotropy) causes the dependence of the elastic modulus on the magnetization of the material (ΔE-effect [57]), therefore, to reduce the ΔE-effect, it is necessary to reduce the elastic modulus.

The films III have the most complex magnetic anisotropy (Figure 9), which is expressed in the coupling field in the hysteresis loop (the presence of a coupling field holding part of the sample magnetization in one direction) in the as-deposited film and after annealing at 300 and 400°C. Note that the film VI annealed at 600°C has a relative remanence of 0.64, whereas the film III annealed at 400°C has that of 0.07. However, despite such a low remanence, sample III annealed at 400°C does not show strong anisotropy field, and at the same time, the magnetization and demagnetization curves near the coupling field run almost vertically.

The reason for such a coupling may be the antiparallel alignment of magnetizations in alternating layers of a multilayer film [58], and the effect manifests itself both in the case of magnetization in the film plane [59,60], and during magnetization perpendicular to the plane [61]. Note that this effect manifests itself only in the films III, for which the amorphous layer thickness is highest (about 200 nm) and the entire film thickness is lowest (0.8 μm).

The topological coupling field is analytically expressed in the "orange peel" model [62,63] and is equal to Hcf = (21/2λtfe)-1π2l2Msexp(-23/2πtd/λ) [64], where l and λ are amplitude and wavelength of roughness at the interface, respectively (in our case, they are equal to the grain size [65], Table 1); Ms is saturation magnetization (1.8 T); and tfe and td are the thicknesses of the ferromagnetic layer and the non-ferromagnetic layer (the “dead” layer on the surface of the ferromagnetic layer does not exceed 1 nm [66]), respectively. After substitution of values suitable for the films III, we obtain Hcf equal to ~240 A/m, which is close to the observed value of the coupling field (Figure 9).

After annealing at 500 and 600°C, the amorphous layer crystallizes; this leads to an increase in the coercive field, and the anisotropy field increases due to the strengthening of the columnar structure (Section 3.2). The films IV and V also contain an amorphous layer, however, no coupling field is noticeable in their hysteresis loops, probably owing to the several times smaller thickness of the amorphous layer relative to the entire film thickness (Table 1).

Note that perpendicular loops were measured for all the films; however, they are looking about the same and the same ugly. We bring a couple of loops:
